# Predictors of Clavien–Dindo Grade III–IV or Grade V Complications after Metastatic Spinal Tumor Surgery: An Analysis of Sociodemographic, Socioeconomic, Clinical, Oncologic, and Operative Parameters

**DOI:** 10.3390/cancers16152741

**Published:** 2024-08-01

**Authors:** Rafael De la Garza Ramos, Jessica Ryvlin, Ali Haider Bangash, Mousa K. Hamad, Mitchell S. Fourman, John H. Shin, Yaroslav Gelfand, Saikiran Murthy, Reza Yassari

**Affiliations:** 1Spine Research Group, Montefiore Medical Center, Albert Einstein College of Medicine, Bronx, NY 10467, USA; jessica.ryvlin@einsteinmed.org (J.R.); alihaider2022-010@stmu.edu.pk (A.H.B.); hamadmousa@gmail.com (M.K.H.); mfourman@montefiore.org (M.S.F.); ygelfand@montefiore.org (Y.G.); samurthy@montefiore.org (S.M.); ryassari@montefiore.org (R.Y.); 2Department of Neurological Surgery, Montefiore Medical Center, Albert Einstein College of Medicine, Bronx, NY 10467, USA; 3Department of Orthopedic Surgery, Montefiore Medical Center, Albert Einstein College of Medicine, Bronx, NY 10467, USA; 4Department of Neurological Surgery, Massachusetts General Hospital, Harvard Medical School, Boston, MA 02115, USA; shin.john@mgh.harvard.edu

**Keywords:** metastatic spinal tumors, outcomes, sociodemographic, socioeconomic, complications, mortality, Clavien–Dindo, vulnerability, modified Bauer score, health disparities

## Abstract

**Simple Summary:**

This research aims to understand the influence of sociodemographic, socioeconomic, clinical, oncologic, and operative parameters on a patient’s risk of major complications or death within 30 days after surgery for spinal tumors that have spread from other parts of the body. The researchers looked at data from 165 patients who had this surgery at a major cancer center between 2012–2023. The findings suggest that a patient’s background factors do not impact their short-term surgical outcomes. Instead, factors like the patient’s overall health, spinal cord compression severity, and nutritional status seem more important. The research community may find these findings helpful in optimizing outcomes for patients undergoing complex spinal tumor surgeries.

**Abstract:**

The rate of major complications and 30-day mortality after surgery for metastatic spinal tumors is relatively high. While most studies have focused on baseline comorbid conditions and operative parameters as risk factors, there is limited data on the influence of other parameters such as sociodemographic or socioeconomic data on outcomes. We retrospectively analyzed data from 165 patients who underwent surgery for spinal metastases between 2012–2023. The primary outcome was development of major complications (i.e., Clavien–Dindo Grade III–IV complications), and the secondary outcome was 30-day mortality (i.e., Clavien–Dindo Grade V complications). An exploratory data analysis that included sociodemographic, socioeconomic, clinical, oncologic, and operative parameters was performed. Following multivariable analysis, independent predictors of Clavien–Dindo Grade III–IV complications were Frankel Grade A–C, lower modified Bauer score, and lower Prognostic Nutritional Index. Independent predictors of Clavien–Dindo Grade V complications) were lung primary cancer, lower modified Bauer score, lower Prognostic Nutritional Index, and use of internal fixation. No sociodemographic or socioeconomic factor was associated with either outcome. Sociodemographic and socioeconomic factors did not impact short-term surgical outcomes for metastatic spinal tumor patients in this study. Optimization of modifiable factors like nutritional status may be more important in improving outcomes in this complex patient population.

## 1. Introduction

Complication rates after oncologic surgery for metastatic spine disease are high [1,2,3,4,5,6,7]. Major adverse events such as unplanned return to the operating room or complications requiring management in an intensive care unit can all have a detrimental impact on a patient’s postoperative course, potentially delaying chemotherapy or radiation treatment. Major complications, including perioperative mortality, are currently estimated to affect up to 35% of patients undergoing surgery, and the rate can be influenced by several risk factors [1,2,3,8,9].

Factors reported in the literature include older age, multilevel metastases, baseline performance status, frailty, malnutrition, and others [1,8,9,10,11]. Some studies have also found that race and insurance status affect the overall complication and non-routine discharge rate after spinal tumor surgery [12,13], but the association between multiple sociodemographic and socioeconomic factors with major complication rates or 30-day mortality has been understudied. In fact, most studies currently omit important covariates such as social vulnerability or primary language, which may play a role in the short-term morbidity and mortality of these patients and procedures.

Thus, the purpose of our study was to perform an exploratory data analysis into the association, if any, of sociodemographic, socioeconomic, clinical, oncologic, and operative parameters with the occurrence of Clavien–Dindo Grade III–IV or Grade V complications.

## 2. Materials and Methods

### 2.1. Study Design and Setting

This research investigation, which was carried out at an urban teaching hospital in a large metropolitan region, was retrospective and single-center. The hospital is affiliated to a cancer center that has been recognized by the National Cancer Institute. In July 2023, a query was made to our neurosurgical spine operative database to identify our potential research group.

### 2.2. Patients

We surgically managed 168 patients suffering from spinal metastases, spinal cord compression, pathologic vertebral compression fractures, and/or spinal mechanical instability between April 2012 and February 2023. Patients who met the following criteria were included: they had to be tracked until death or for at least 30 days after surgery, and they had to have comprehensive sociodemographic, socioeconomic, clinical, oncologic, operative, and follow-up data. These criteria resulted in the exclusion of 1% (2) patients for incomplete data and <1% (1) patient for loss prior to the minimum study follow-up time, leaving 98% (165 of 168) of the patients for final analysis. Among our analytic sample, surgical indications included metastatic spinal cord compression in 86% (142) of patients, pathologic vertebral compression fractures in 50% (83) of patients, 29% (48) of patients had unstable lesions, and 64% (106) of patients had potentially unstable lesions as identified by the Spinal Instability Neoplastic Score (SINS). Surgical decompression was performed for patients with myelopathy, motor weakness, or inability to ambulate if they presented within 48 h of impairment; internal fixation was performed in patients who were at risk for iatrogenic instability from surgical decompression or patients with a SINS indicative of potential instability or instability.

### 2.3. Variables

Collected sociodemographic and socioeconomic data included age, sex (male vs. female), self-reported race: White, Black, Hispanic or Latino, or other (Middle Eastern, Asian, or Southeast Asian), primary language (English versus non-English), primary insurance status at the time of surgery (Medicare, Medicaid, or private), year of surgery, and SVI score and SVI subtheme scores (socioeconomic status, household composition and disability, minority status and language, and housing type and transportation). The SVI scores were obtained from the Centers for Disease Control and Prevention’s website (https://svi.cdc.gov/map.html, (accessed on 1 September 2023)); the year immediately preceding the date of surgery was used as well as patient’s domicile at the time of surgery. These scores range from 0–1, with a higher value being indicative of higher vulnerability or deprivation.

Collected clinical, oncological, and operative data included Eastern Cooperative Oncology Group (ECOG) performance status within 30 days of surgery, body mass index (BMI), American Society of Anesthesiologists class, Frankel grade at presentation (D–E vs. A–C), primary cancer (lung, breast, prostate, kidney, thyroid, colorectal, hematologic, or other), modified Bauer score, preoperative Prognostic Nutritional Index (PNI), de novo cancer diagnosis, SINS, emergency-type procedure (performed within 24 h of admission), use of internal fixation, open procedure, number of instrumented vertebrae, and use of transpedicular decompression. The scales used are summarized and referenced in the Appendix A.

### 2.4. Primary and Secondary Study Endpoints

The primary endpoint was development of at least one Clavien–Dindo Grade III or IV complication within 30 days of surgery [10]. These are considered major surgical complications and include complications requiring surgical, endoscopic, or radiological intervention (such as epidural hematoma or wound infection requiring revision surgery), or life-threatening complications requiring intensive care unit admission (such as adult respiratory distress syndrome or unplanned intubation) [10]. The secondary endpoint was development of a Clavien–Dindo Grade V complication within 30 days of surgery, which is defined as 30-day mortality.

### 2.5. Statistical Analysis

All analyses were performed in Stata 16 IC (StataCorp, College Station, TX, USA). An initial data exploration was carried out where distribution of data was assessed using histograms and the Kolmogorov–Smirnov test. A univariable logistic regression analysis was carried out with Clavien–Dindo Grade III or IV complications as the main dependent variable; for the secondary outcome the same analysis was performed with Clavien–Dindo Grade IV complications as the main dependent variable. The included independent variables were age, sex, race (White, Black, Hispanic or Latino, or other), non-English primary language, Medicare primary insurance, Medicaid primary insurance, private primary insurance, year of surgery (2012–2017 or 2018–2023), overall SVI score, socioeconomic status score, household composition and disability score, racial or ethnic group and language score, housing type and transportation score, ECOG performance status, BMI, ASA class, Frankel grade A–C, primary cancer (lung, breast, prostate, kidney, thyroid, colorectal, hematologic, or other), modified Bauer score, preoperative PNI, de novo cancer diagnosis, SINS, emergency-type procedure, use of internal fixation, open procedure, number of instrumented vertebrae, and use of transpedicular decompression. All factors with a *p* value less than 0.10 were then included in a multivariable stepwise logistic regression analysis with backward elimination. Results are presented as odds ratios (OR) with 95% confidence intervals (CI). Statistical significance was defined as a *p* value less than 0.05.

## 3. Results

### 3.1. Patients’ Baseline Data

A total of 165 patients were included in this study (Table 1). The median age of our study population was 63 years (interquartile range [IQR] 54 to 70), and 62% (102 of 165) of patients were men. The race distribution was 18% (*n* = 29) White, 45% (*n* = 74) Black, 30% (*n* = 49) Hispanic or Latino, and 8% (*n* = 13) other. Non-English was the primary language in 21% (*n* = 35) of patients. The insurance distribution was as follows: Medicare: 38% (*n* = 63), Medicaid: 36% (*n* = 60), and private insurance: 21% (*n* = 35). The median SVI score was 89.8 (IQR 72.6 to 98.0). The median ECOG performance status was 2 (IQR 1 to 3) and the median ASA Class was 3 (IQR 3 to 3) (Table 2). From the entire group, 21% (35 of 165) of patients presented with Frankel Grade A to C and 32% (*n* = 52) with complete inability to walk. The median modified Bauer score was 2 (1 to 3) and the mean PNI was 42.5 ± 7.7.

### 3.2. Univariable Analysis of Factors Associated with Clavien–Dindo Grade III–IV Complications and Grade V Complications

From the total study group, 26% (43 of 165) of patients developed at least one Clavien–Dindo Grade III or IV complication. These complications included unplanned return to the operating room (*n* = 12), sepsis (*n* = 11), pulmonary embolism (*n* = 9), unplanned intubation (*n* = 7), adult respiratory distress syndrome (*n* = 5), meningitis (*n* = 3), and stroke (*n* = 2). The rate of Clavien–Dindo Grade V complications was 8.5% (14 of 165). The crude rates of these complications as well as univariable analysis are summarized in Table 2. No sociodemographic or socioeconomic factor was significantly associated with these outcomes on univariable analysis.

### 3.3. Multivariable Analysis of Factors Associated with Clavien–Dindo Grade III–IV Complications and Grade V Complications

After controlling for ECOG performance status, Frankel Grade A–C, lung cancer, modified Bauer score, PNI, SINS, and emergency procedures (Table 3), independent factors associated with development of at least one Clavien–Dindo Grade III or IV complication were Frankel Grade A–C (OR 6.2, 95% CI 2.4 to 15.5; *p* < 0.001), the modified Bauer score (OR 0.5, 95% CI 0.3 to 0.91; *p* = 0.02), and the PNI (OR 0.9, 95% CI 0.8 to 0.9; *p* = 0.02).

After controlling for other race, ECOG performance status, BMI, ASA class, Frankel Grade A–C, lung cancer, modified Bauer score, PNI, and internal fixation, independent factors associated with development of a Clavien–Dindo Grade V complication were lung primary cancer (OR 5.2, 95% CI 1.1 to 24.5; *p* = 0.04), modified Bauer score (OR 0.4; 95% CI 0.2 to 0.9; *p* = 0.03), the PNI (OR 0.9, 95% CI 0.8 to 0.9; *p* = 0.04), and use of internal fixation (OR 0.1; 95% CI, 0.1 to 0.4; *p* = 0.01) (Table 3).

## 4. Discussion

The rate of major complications including perioperative mortality after metastatic spinal tumor surgery is relatively high, estimated at 16–34% [1,2,3,4]. While many different factors such as age, multilevel metastases, baseline performance status, frailty, and malnutrition have been shown to be associated with adverse events [1,10,11], data also accounting for other important sociodemographic or socioeconomic data such as race, primary language, and social vulnerability, among others, are limited. In general surgical oncology, disparities in outcomes have predominantly affected minority patients and patients with low socioeconomic status [14]. Black and African-American patients have been shown to have higher risk of adverse events and perioperative mortality across multiple studies [15,16,17]. Similarly, socially vulnerable patients and patients without private insurance tend to have worse outcomes [18].

The present study sought to examine a cohort of surgical patients and perform an exploratory analysis accounting for many different variables, including sociodemographic, socioeconomic, clinical, oncological, and operative factors and their association with major perioperative morbidity. We found that no single sociodemographic or socioeconomic factor was associated with the development of Clavien–Dindo Grade III or IV complications (major complications) after metastatic spinal tumor surgery. On the other hand, the only clinical factors associated with this endpoint were a preoperative Frankel Grade A–C, the modified Bauer score, and the PNI. Although several other studies have found that race is associated with a higher likelihood of overall complications after oncologic spine surgery [19], a study looking at both minor and major complications found that race was associated with minor, but not major complications, similar to our findings [12]. An institutional series of 328 patients found that race, insurance, and income were not associated with postoperative complications [13]. Likewise, a study examining the impact of insurance status on in-hospital mortality and complications found that after adjusting for acuity of presentation, socioeconomic status, hospital bed size, and hospital teaching status, no difference in outcomes were found [20]. These results suggest that perioperative complications are more likely the result of baseline patient characteristics such as the neurologic exam, extent of disease, and nutritional status, among others.

When examining Clavien–Dindo Grade V complications, we also found that no sociodemographic or socioeconomic factor was associated with this outcome. Short-term mortality, particularly within 30 days, is usually more related to a patient’s preoperative functional status, nutritional/inflammatory status, and the development of any perioperative adverse event rather than the primary tumor pathology [8,21,22]. Our findings are consistent, however, with other studies that found that factors such as race, insurance, or social vulnerability were not associated with post-treatment survival on multivariable analysis [19,23,24].

Independent factors associated with both outcomes studied here included the modified Bauer score and the PNI. The former is a composite score that includes primary cancer type as well as extent of disease (assessed by the presence of visceral metastasis and solitary vs. multiple metastatic lesions) and indicates that a patient’s baseline extent of disease is perhaps more predictive of the postoperative course as opposed to a particular sociodemographic or socioeconomic parameter [25,26]. Similarly, the PNI is a measure of the nutritional-inflammatory status of a patient and has been found to be associated with outcomes in oncologic spine surgery including postoperative survival and complication occurrence [11,27,28]. While formal research in patients with metastatic spine disease is lacking, nutritional supplementation of cancer patients is a potentially modifiable risk factor that could impact the surgical recovery and tolerability of adjuvant therapy [29,30].

There are several limitations to the study, particularly selection bias. Our study is a single-center experience so results may not be entirely generalizable to other populations. Our study group also consisted of mostly minority and socially vulnerable patients which may not represent other study populations. The lack of an association between sociodemographic or socioeconomic parameters may also be the result of sample size and unmeasured covariates, among others. Nonetheless, over 35 multi-dimensional variables were analyzed in a detailed univariable and multivariable analysis, as opposed to previous studies focused more on comorbidities and surgical parameters as predictors of major perioperative morbidity.

## 5. Conclusions

The present study sought to examine the association between different multi-dimensional parameters and the development of major complications including 30-day mortality after surgery for metastatic spinal tumors. We found that no sociodemographic or socioeconomic factor predicted these events. On the other hand, preoperative neurologic status, the modified Bauer score, and nutritional status, among others, may be more likely responsible for short-term outcome in this challenging patient population. These findings may prove useful for preoperative risk stratification and future research into potential optimization strategies.

## Figures and Tables

**Table 1 cancers-16-02741-t001:** Sociodemographic, socioeconomic, clinical, oncologic, and operative data of 165 patients.

Parameter	Value
Age in years, median (IQR)	63 (54 to 70)
Male, % (*n*)	61.8 (102)
Race, % (*n*)	
White	17.6 (29)
Black	44.9 (74)
Hispanic/Latino	29.7 (49)
Other	7.8 (13)
Primary language, % (*n*)	
English	78.8 (130)
Non-English	21.2 (35)
Primary insurance, % (*n*)	
Medicare	38.2 (63)
Medicaid	36.4 (60)
Private	21.2 (35)
Year of surgery	
2012–2017	49.7 (82)
2018–2023	50.3 (83)
Social Vulnerability Index, median (IQR)	89.8 (72.6 to 98.0)
SVI subthemes, median (IQR)	
Socioeconomic status	80.4 (56.8 to 93.9)
Household composition and disability	70.1 (44.9 to 86.4)
Minority status and language	91.5 (85.0 to 97.6)
Housing type and transportation	87.6 (73.7 to 95.9)
ECOG performance status, % (*n*)	
0	5.6 (9)
1	36.4 (60)
2	30.3 (50)
3	23.6 (39)
4	4.2 (7)
BMI in kg/m^2^, mean ± SD	26.6 ± 5.6
ASA Class, median (IQR)	3 (3 to 3)
Frankel Grade, % (*n*)	
Frankel D–E	78.9 (130)
Frankel A–C	21.1 (35)
Primary cancer, % (*n*)	
Breast	15.2 (25)
Lung	15.8 (26)
Prostate	20.0 (33)
Colorectal	4.9 (8)
Kidney	4.9 (8)
Hematologic	20.0 (33)
Other	17.8 (29)
Modified Bauer score, median (IQR)	2 (1 to 3)
Prognostic Nutritional Index, mean ± SD	42.5 ± 7.7
De novo cancer diagnosis, % (*n*)	37.0 (61)
SINS, median (IQR)	11 (8 to 13)
Emergency procedure, % (*n*)	26.0 (43)
Internal fixation	88.5 (146)
Open procedure	73.3 (121)
Number of instrumented levels, median (IQR)	4 (4 to 6)
Transpedicular decompression	55.8 (92)

ECOG = Eastern Cooperative Oncology Group; SINS = Spinal Instability Neoplastic Score.

**Table 2 cancers-16-02741-t002:** Outcomes by independent risk factor (univariable analysis).

Parameter	Clavien–Dindo Grade III–IV Complication	Odds Ratio with 95% CI	Clavien–Dindo Grade V Complication	Odds Ratio with 9% CI
Increasing Age		0.9 (0.9 to 1.1) *p* = 0.52		1.0 (0.9 to 1.1) *p* = 0.21
Male vs. Female	27% vs. 25%	1.1 (0.5 to 2.2) *p* = 0.88	9% vs. 8%	1.1 (0.4 to 3.5) *p* = 0.84
White vs. Not-White	28% vs. 26%	1.1 (0.5 to 2.7) *p* = 0.84	10% vs. 8%	1.3 (0.3 to 5.0) *p* = 0.69
Black vs. Not-Black	27% vs. 25%	1.1 (0.5 to 2.2) *p* = 0.80	5% vs. 11%	0.5 (0.2 to 1.5) *p* = 0.21
Hispanic/Latino vs. Not Hispanic/Latino	20% vs. 29%	0.6 (0.3 to 1.4) *p* = 0.29	8% vs. 9%	0.9 (0.3 to 2.2) *p* = 0.92
Other race vs. Not Other race	38% vs. 25%	1.9 (0.6 to 6.1) *p* = 0.30	23% vs. 7%	3.9 (0.9 to 16.1) *p* = 0.07 *
Non-English primary language vs. English primary language	17% vs. 28%	0.5 (0.2 to 1.4) *p* = 0.18	9% vs. 8%	1.0 (0.3 to 3.9) *p* = 0.98
Medicare insurance vs. no	24% vs. 27%	0.8 (0.4 to 1.7) *p* = 0.61	8% vs. 9%	0.9 (0.3 to 2.8) *p* = 0.84
Medicaid Insurance vs. no	25% vs. 27%	0.9 (0.4 to 1.9) *p* = 0.82	10% vs. 8%	1.3 (0.4 to 4.1) *p* = 0.60
Private insurance vs. no	29% vs. 25%	1.2 (0.5 to 2.7) *p* = 0.70	6% vs. 9%	0.6 (0.1 to 2.8) *p* = 0.51
Year of surgery 2012–2017 vs. 2018–2023	28% vs. 24%	1.2 (0.6 to 2.4) *p* = 0.627	6% vs. 11%	0.7 (0.4 to 1.3) *p* = 0.31
Increasing SVI		0.8 (0.2 to 3.9) *p* = 0.80		1.1 (0.1 to 13.3) *p* = 0.96
Increasing socioeconomic status vulnerability		0.9 (0.3 to 3.5) *p* = 0.93		1.6 (0.2 to 13.8) *p* = 0.69
Increasing household composition and disability vulnerability		1.1 (0.3 to 4.3) *p* = 0.92		5.0 (0.4 to 58.4) *p* = 0.20
Increasing minority status and language vulnerability		0.6 (0.1 to 5.1) *p* = 0.64		0.5 (0.1 to 12.6) *p* = 0.69
Increasing housing type and transportation vulnerability		0.9 (0.2 to 5.4) *p* = 0.92		0.9 (0.1 to 14.9) *p* = 0.95
Increasing ECOG performance status		1.8 (1.2 to 2.6) *p* = 0.01 *		2.3 (1.3 to 4.2) *p* = 0.01 *
Increasing BMI		1.0 (0.9 to 1.1) *p* = 0.95		0.8 (0.7 to 0.9) *p* = 0.01 *
Increasing ASA Class		1.4 (0.8 to 2.5) *p* = 0.31		2.2 (0.9 to 5.7) *p* = 0.09 *
Frankel Grade A–C vs. D–E	51% vs. 19%	4.4 (2.0 to 9.8) *p* < 0.001 *	17% vs. 6%	3.2 (1.1 to 9.8) *p* = 0.05 *
Breast cancer vs. no breast cancer	16% vs. 28%	0.5 (0.2 to 1.5) *p* = 0.22	4% vs. 9%	0.4 (0.1 to 3.3) *p* = 0.40
Lung cancer vs. no lung cancer	42% vs. 23%	2.5 (1.1 to 5.9) *p* = 0.04	31% vs. 4%	9.9 (3.1 to 31.7) *p* < 0.001 *
Prostate cancer vs. no prostate cancer	18% vs. 28%	0.6 (0.2 to 1.5) *p* = 0.25	0% vs. 11%	Omitted
Colorectal cancer vs. no colorectal cancer	25% vs. 26%	0.9 (0.2 to 4.9) *p* = 0.94	13% vs. 8%	1.6 (0.2 to 13.9) *p* = 0.68
Kidney cancer vs. no kidney cancer	38% vs. 25%	1.8 (0.4 to 7.7) *p* = 0.46	13% vs. 8%	1.6 (0.2 to 13.9) *p* = 0.68
Hematologic cancer vs. no hematologic cancer	21% vs. 27%	0.7 (0.3 to 1.8) *p* = 0.48	6% vs. 9%	0.6 (0.2 to 3.0) *p* = 0.58
Other cancer vs. no other cancer	34% vs. 24%	1.6 (0.7 to 3.9) *p* = 0.26	4% vs. 10%	0.3 (0.1 to 2.7) *p* = 0.31
Increasing modified Bauer score		0.5 (0.3 to 0.8) *p* = 0.001 *		0.3 (0.2 to 0.6) *p* < 0.001 *
Increasing PNI		0.9 (0.8 to 0.9) *p* = 0.005 *		0.9 (0.8 to 0.9) *p* = 0.02 *
De novo cancer diagnosis vs. no	26% vs. 26%	1.0 (0.5 to 2.1) *p* = 0.97	10% vs. 8%	1.3 (0.4 to 3.9) *p* = 0.63
Increasing SINS		0.9 (0.8 to 0.9) *p* = 0.02 *		1.0 (0.8 to 1.2) *p* = 0.97
Emergency procedure vs. no	40% vs. 21%	2.4 (1.1 to 5.1) *p* = 0.02 *	12% vs. 7%	1.7 (0.5 to 5.2) *p* = 0.39
Internal fixation vs. no	25% vs. 37%	0.6 (0.2 to 1.5) *p* = 0.26	9% vs. 21%	0.3 (0.1 to 0.9) *p* = 0.05 *
Open procedure vs. no	29% vs. 18%	1.8 (0.8 to 4.3) *p* = 0.17	9% vs. 7%	1.4 (0.4 to 5.2) *p* = 0.64
Increasing number of instrumented levels		1.0 (0.8 to 1.3) *p* = 0.72		0.8 (0.5 to 1.2) *p* = 0.25
Transpedicular decompression vs. no	22% vs. 32%	0.6 (0.3 to 1.2) *p* = 0.16	5% vs. 12%	0.4 (0.1 to 1.3) *p* = 0.12

SVI: social vulnerability index; ECOG: Eastern Cooperative Oncology Group; BMI: body mass index; ASA: American Society of Anesthesiologists; PNI: Prognostic Nutritional Index; SINS: spinal instability neoplastic score; * Included in the multivariable model.

**Table 3 cancers-16-02741-t003:** Stepwise multivariable logistic regression of factors associated with Clavien–Dindo Grade III–IV Complications or Grade V Complications.

Clavien–Dindo Grave III–IV Complications			
**Parameter**	**OR**	**95% CI**	***p* Value**
ECOG performance status	1.2	0.8 to 2.0	0.37
Frankel Grade A–C	6.2	2.4 to 15.5	<0.001 *
Lung	0.8	0.2 to 2.8	0.76
Modified Bauer score	0.6	0.4 to 0.9	0.01 *
Prognostic Nutritional Index	0.9	0.8 to 0.9	0.01 *
SINS	0.9	0.8 to 1.0	0.08
Emergency procedure	2.2	0.8 to 6.5	0.15
**Clavien–Dindo Grave V Complications**			
**Parameter**	**OR**	**95% CI**	***p* Value**
Other race	6.4	0.7 to 57.8	0.10
ECOG performance status	1.2	0.8 to 2.0	0.37
BMI	0.9	0.7 to 1.0	0.15
ASA Class	1.6	0.4 to 5.8	0.48
Frankel Grade A–C	3.5	0.6 to 20.5	0.17
Lung	5.2	1.1 to 24.5	0.04 *
Modified Bauer score	0.4	0.2 to 0.9	0.03 *
Prognostic Nutritional Index	0.9	0.8 to 0.9	0.04 *
Internal fixation	0.1	0.1 to 0.4	0.01 *

ECOG: Eastern Cooperative Oncology Group; BMI: body mass index; ASA: American Society of Anesthesiologists; SINS: spinal instability neoplastic score; * statistically significant finding.

## Data Availability

The data presented in this study are available on request from the corresponding author due to privacy concerns.

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
