# Peer review of "Predictors of Clavien–Dindo Grade III–IV or Grade V Complications after Metastatic Spinal Tumor Surgery: An Analysis of Sociodemographic, Socioeconomic, Clinical, Oncologic, and Operative Parameters"

_cancers, 2024, doi:10.3390/cancers16152741_

Round 1

Reviewer 1 Report

Comments and Suggestions for Authors

This study aim was to understand whether a patient's sociodemographic or socioeconomic factors influence their risk of major complications or death within 30 days after surgery 6 for spinal tumors that have spread from other parts of the body.

The two stated aims at the end of the introduction were:

1. Are sociodemographic or socioeconomic factors associated with major (Clavien-Dindo Grade III-IV) complications?

2. Are sociodemographic or socioeconomic factors associated with 30-day mortality (Clavien-Dindo Grade V) complications?

Variables:

The title and stated focus of the paper is misleading based on the variables included in the analysis and the analysis type. As stated this is an exploratory analysis but it is not just focused on demographic variables. The scope of the paper contained both SES as well as clinical baseline variables (ie ASA, etc) so the stated focus on SES is misleading and unnecessary. If the analysis included certain demographic variables (ie race, insurance status) and there was a comparison between black and white patients or these SES were assessed while accounting for clinically relevant or statistical confounding  variables then the title would make more sense. Why was year of surgery not assessed or accounted for? 

Outcomes:

I appreciate the structure used for the manuscript in identified objectives but again I feel like the stated aims are a little misleading. Isn't mortality a component of the  Clavien-Dindo Grade V complication? Please rewrite this section to make this more clear. 

Statistical Analysis

There is an error or typo (line 113) as the listed variables are independent not dependent variables. What were the parameters for inclusion in a final model? Please state this you list what is defined as significant but not the threshold for inclusion in a model. Additionally if you were focusing/assessing SES variables you would have needed to include (ie force) their inclusion while adjusting for covariates based on your threshold. Please change. This is a multivariable not a multivariate analysis PMID: 23153131.

Results:

What is the difference between Table 1 and 2? I don't understand the purpose of separating out the SES factors as the analysis was not conducted that way and it's misleading. Please list all baseline factors in table 1 and organize by type. Table 2 should include the N of your your primary and secondary outcomes at a minimum by independent risk factor. Please list this in the results section and in table 2. You can do side by side of the overall score and the sub mortality score with the univariate ORs next to this.

Table 3 can be the results of the multivariable stepwise logistic regression for overall score and then the mortality component. In the tables the way the variables are listed makes no sense. Did you assess 'other race' as a binary y/no? If not then what is listed is misleading. List ALL the levels of class variables not just the ones that are significant. In your methods section list what is the reference variable for every ordinal or nominal variable since this will often heavily influence significance of individual components of that variable.

Please remove all statements that say care is equitable and there is no difference. First of all you didn't assess that and second a single study could not determine that. This is a risk factor analysis or an exploratory analysis that includes clinical and baseline variables. You failed to detect a difference which is how it should be written which could be the case for many reasons including misclassification, sample size/outcomes, unmeasured confounding, etc. Please be thorough in your limitations section. 

Author Response

Reviewer 2

This study aim was to understand whether a patient's sociodemographic or socioeconomic factors influence their risk of major complications or death within 30 days after surgery 6 for spinal tumors that have spread from other parts of the body.

The two stated aims at the end of the introduction were:

  1. Are sociodemographic or socioeconomic factors associated with major (Clavien-Dindo Grade III-IV) complications?
  2. Are sociodemographic or socioeconomic factors associated with 30-day mortality (Clavien-Dindo Grade V) complications?

Variables:

The title and stated focus of the paper is misleading based on the variables included in the analysis and the analysis type. As stated this is an exploratory analysis but it is not just focused on demographic variables. The scope of the paper contained both SES as well as clinical baseline variables (ie ASA, etc) so the stated focus on SES is misleading and unnecessary. If the analysis included certain demographic variables (ie race, insurance status) and there was a comparison between black and white patients or these SES were assessed while accounting for clinically relevant or statistical confounding  variables then the title would make more sense. Why was year of surgery not assessed or accounted for? 

Thank you for this comment and suggestion. We have revised this. We also added the year of surgery as a variable, which was found not to be significantly associated with any of the two examined outcomes.

Outcomes:

I appreciate the structure used for the manuscript in identified objectives but again I feel like the stated aims are a little misleading. Isn't mortality a component of the  Clavien-Dindo Grade V complication? Please rewrite this section to make this more clear. 

Thank you for this suggestion. We have revised this.

“The secondary outcome was development of a Clavien-Dindo Grade V complication within 30 days of surgery, which is defined as 30-day mortality.”

Statistical Analysis

There is an error or typo (line 113) as the listed variables are independent not dependent variables. What were the parameters for inclusion in a final model? Please state this you list what is defined as significant but not the threshold for inclusion in a model. Additionally if you were focusing/assessing SES variables you would have needed to include (ie force) their inclusion while adjusting for covariates based on your threshold. Please change. This is a multivariable not a multivariate analysis PMID: 23153131.

Thank you. We have revised this thoroughly and made corrections.

Results:

What is the difference between Table 1 and 2? I don't understand the purpose of separating out the SES factors as the analysis was not conducted that way and it's misleading. Please list all baseline factors in table 1 and organize by type.

Thank you for this suggestion. We have merged Table 1 and Table 2 into a single Table (1).

Table 2 should include the N of your primary and secondary outcomes at a minimum by independent risk factor. Please list this in the results section and in table 2. You can do side by side of the overall score and the sub mortality score with the univariate ORs next to this.

Thank you for this suggestion. We have revised Table 2.

Table 3 can be the results of the multivariable stepwise logistic regression for overall score and then the mortality component. In the tables the way the variables are listed makes no sense. Did you assess 'other race' as a binary y/no? If not then what is listed is misleading. List ALL the levels of class variables not just the ones that are significant. In your methods section list what is the reference variable for every ordinal or nominal variable since this will often heavily influence significance of individual components of that variable.

Thank you for this suggestion. We have revised Table 3.

Please remove all statements that say care is equitable and there is no difference. First of all you didn't assess that and second a single study could not determine that. This is a risk factor analysis or an exploratory analysis that includes clinical and baseline variables. You failed to detect a difference which is how it should be written which could be the case for many reasons including misclassification, sample size/outcomes, unmeasured confounding, etc. Please be thorough in your limitations section. 

Thank you for this comment and suggestion. We have revised our entire manuscript including our discussion and limitations sections.

Reviewer 2 Report

Comments and Suggestions for Authors

Dear authors,

Your study is interesting and generally well-conducted. My comments are minor and intended to enhance the readability and clarity of data presentation:

  1. Please explain why the period between April 2012 and February 2023 was selected.
  2. Clarify if your hospital has witnessed substantial changes in approaches to managing patients with spinal metastases, spinal cord compression, pathological vertebral compression fractures, and/or spinal mechanical instability during the study period.
  3. Consider adding a subsection debriefing all scales used, and reference them appropriately.
  4. The "statistical analysis" subsection starts with a description of regression analysis. Consider adding a passage describing approaches to data analysis used. For example, you report continuous variables as the median and IQR. Explain how you checked your data for normality of distribution.
  5. Finally, the Ethics Statement appears to be missing.

Author Response

Dear authors,

Your study is interesting and generally well-conducted. My comments are minor and intended to enhance the readability and clarity of data presentation:

  1. Please explain why the period between April 2012 and February 2023 was selected.

Thank you for this question. These are the years that we had data available for in our operative log.

  1. Clarify if your hospital has witnessed substantial changes in approaches to managing patients with spinal metastases, spinal cord compression, pathological vertebral compression fractures, and/or spinal mechanical instability during the study period.

No significant changes have occurred throughout the years.

  1. Consider adding a subsection debriefing all scales used, and reference them appropriately.

Thank you for this suggestion. We have added supplementary files describing all of the scales used with references.

  1. The "statistical analysis" subsection starts with a description of regression analysis. Consider adding a passage describing approaches to data analysis used. For example, you report continuous variables as the median and IQR. Explain how you checked your data for normality of distribution.

Thank you for this suggestion. We have corrected our methods section to reflect this.

“Distribution of data was assessed using histograms and the Kolmogorov-Smirnov test.”

  1. Finally, the Ethics Statement appears to be missing.

Thank you for this observation. The ethics statement appears at the end of our manuscript.

Round 2

Reviewer 1 Report

Comments and Suggestions for Authors

Thank you for providing a thorough revision of the manuscript. The updates in this version are sufficient for publication.